# The Power of Vitamin D: Is the Future in Precision Nutrition through Personalized Supplementation Plans?

**DOI:** 10.3390/nu16081176

**Published:** 2024-04-15

**Authors:** Mladen Mavar, Tamara Sorić, Ena Bagarić, Ana Sarić, Marijana Matek Sarić

**Affiliations:** 1Psychiatric Hospital Ugljan, Otočkih Dragovoljaca 42, 23275 Ugljan, Croatia; ravnatelj@pbu.hr; 2Almagea Ltd., Ulica Julija Knifera 4, 10020 Zagreb, Croatia; ena.bagaric@gmail.com; 3School of Medicine, Catholic University of Croatia, Ilica 242, 10000 Zagreb, Croatia; asaric1@unicath.hr; 4Department of Health Studies, University of Zadar, Splitska 1, 23000 Zadar, Croatia; marsaric@unizd.hr

**Keywords:** deficiency, precision nutrition, supplementation, vitamin D

## Abstract

In the last few decades, vitamin D has undeniably been one of the most studied nutrients. Despite our ability to produce vitamin D through sunlight exposure, its presence in several natural food sources and fortified foods, and its widespread availability as a dietary supplement, vitamin D deficiency is a serious public health problem, affecting nearly 50% of the global population. Low serum levels of vitamin D are being associated with increased susceptibility to numerous health conditions, including respiratory infections, mental health, autoimmune diseases, and different cancer types. Although the association between vitamin D status and health is well-established, the exact beneficial effects of vitamin D are still inconclusive and indefinite, especially when considering the prevention and treatment of different health conditions and the determination of an appropriate dosage to exert those beneficial effects in various population groups. Therefore, further research is needed. With constant improvements in our understanding of individual variations in vitamin D metabolism and requirements, in the future, precision nutrition and personalized supplementation plans could prove beneficial.

## 1. Introduction

Vitamin D is a steroid hormone that is vitally important for all vertebrates, including humans. Maintaining normal serum levels of calcium and phosphate is critically dependent on vitamin D. Vitamin D thus contributes to the well-being of the human body by promoting a strong skeletal system, facilitating muscle movement, supporting the immune system, and optimizing cellular functions throughout the body [1,2].

Vitamin D is a fat-soluble vitamin found naturally in a small selection of foods or artificially added to other foods (i.e., foods fortified with vitamin D). It is also widely available as a dietary supplement. Additionally, our body can produce vitamin D through skin exposure to sunlight, which represents the main source of vitamin D for humans.

When it comes to the origin of its name, it is interesting that it is not a chemical term, but a name adopted by nutritionists exactly 100 years ago and defined as ‘a substance with anti-rachitic properties that will cure rickets’ [1,2,3].

The discovery of vitamin D is attributed to several scientists that worked independently. In 1919, Sir Edward Mellanby discovered that a component in cod liver oil was crucial for the prevention of rickets in dogs, although he did not identify the exact substance responsible. In 1922, Elmer McCollum and Marguerite Davis discovered vitamin D while conducting experiments on rats [4,5,6,7,8]. Later the same year, Harry Steenbock discovered that the irradiation of food could produce vitamin D and he developed a method of fortifying milk with vitamin D [9,10]. Therefore, the discovery of vitamin D is credited to multiple scientists who formed the basis of our current understanding of this essential nutrient. 

The research interest in vitamin D, its properties, and recommendations for its intake is still not waning and we could even say it has increased drastically in the last few years, making vitamin D one of the most studied nutrients. 

The present review provides a concise exploration of vitamin D metabolism, investigates the benefits of maintaining the adequate serum vitamin D levels in order to preserve the health of individuals, and assesses existing literature regarding recommendations for vitamin D supplementation. Furthermore, it discusses future perspectives for vitamin D supplementation, including a more personalized approach that could lead to more beneficial results. 

## 2. Vitamin D Metabolism 

Vitamin D enters the bloodstream in the small intestine following ingestion by food or dietary supplements. It is being absorbed through both a simple passive diffusion and a mechanism involving carrier proteins in the intestinal membrane [11]. As vitamin D is a fat-soluble vitamin, its absorption is enhanced by the coexisting presence of fat in the small intestine, and thus it is important to consume vitamin D with a meal that contains some fat. However, a certain proportion of vitamin D can be absorbed even though there is no dietary fat present concurrently. 

It is believed that neither obesity nor aging have an impact on the absorption of vitamin D from the small intestine [11]; however, some of the authors suggest that for patients with obesity, a dose that is several times higher than the one intended for patients with normal body weight is required to both achieve and sustain the normal serum level of vitamin D [12].

On the other hand, in animals that can synthetize cholesterol and some organisms of plant origin, including phytoplankton, exposure to ultraviolet light B (UVB) converts 7-dehydrocholesterol to vitamin D_3_. The described process is called photolysis [3]. To be more specific, after the UVB radiation from the sun enters the skin, it is responsible for the transformation of 7-dehydrocholesterol to previtamin D_3_. When previtamin D_3_ is produced, it is promptly converted to vitamin D_3_ through a thermal reaction [13].

Vitamin D derived from food, as well as the one our skin synthesizes, are actually inactive forms of the vitamin. It is necessary for vitamin D to be converted to its active form to have numerous effects in the human body. The process of activation includes two reactions of hydroxylation. Vitamin D that enters the body through food is absorbed into the lymph through chylomicrons. After vitamin D is converted to its active form, it can attach to vitamin D receptors (VDRs) found on the surfaces of cells expressing the receptors. After the production, vitamin D_3_ is transported to the liver and there it is converted to 25-hydroxyvitamin D [25(OH)D], also known as calcidiol [14]. Calcidiol is the main form of vitamin D circulating in the bloodstream and is used to evaluate one’s vitamin D status. Its half-life in the blood is approximately two to three weeks [15].

Finally, 25(OH)D is transported to the kidneys, where the process of conversion to its active form 1,25-dihydroxyvitamin D [1,25(OH)_2_D] takes place [16,17]. Although kidneys are its major source, different cell types express CYP27B1, a gene that encodes 1α-hydroxylase, an enzyme catalyzing the hydroxylation of 25(OH)D to 1,25(OH)_2_D. Some of the locations of extra-renal vitamin D metabolism are the placenta, monocytes/macrophages, parathyroid gland, ovaries, breasts, colon, lung, and osteoblasts [18].

1,25(OH)_2_D, also called calcitriol, plays a key role in most of vitamin D’s biological functions, such as controlling the absorption of calcium and phosphorus in the intestines and maintaining bone health [19]. Figure 1 provides a concise overview of vitamin D metabolism and its pathway [20].

It is estimated that most of the vitamin D that is naturally present in the human body is derived from sun exposure (approximately 80%), while the remaining 20% is obtained from food sources [21]. The production of vitamin D in the skin is affected by various factors, such as skin color, geographical location, time of the year, time of the day, and the use of sunscreen [22].

The sufficient concentration of 25(OH)D is considered to be around 30 ng/mL (75 nmol/L). The aforementioned dose is the lowest one necessary for the prevention of hyperparathyroidism development. It is also confirmed to be sufficient for effective calcium absorption in the intestines. Vitamin D deficiency is defined as 25(OH)D concentrations < 20 ng/mL (50 nmol/L), and these concentrations are associated with numerous negative health effects, primarily on the musculoskeletal system [12].

## 3. Vitamin D and Its Positive Health Effects

### 3.1. An Inseparable Bond: Vitamin D and Vitamin D Receptors 

By attaching to VDRs, vitamin D serves its purpose. Most physiological processes of 1,25(OH)_2_D are moderated by VDRs. Additionally, VDRs, through its role as a transcription factor, exerts control over several target genes, including those responsible for tight junction proteins like claudin-2, -5, -12, and -15 [23]. Vitamin D, or more specifically, its biologically active form 1,25(OH)_2_D, behaves like a steroid hormone, influencing around 900 different genes or roughly 3% of the human genome, either directly or indirectly [24].

VDRs are expressed on immune cells, which makes them vitamin D targets. By preventing the invasion of harmful bacteria, inhibiting inflammation, and preserving barrier function, the activation of VDRs is crucial in controlling intestinal homeostasis. In addition, VDR signaling has a critical role in controlling other anti-inflammatory and proliferative pathways [23].

When VDRs bind to 1,25(OH)_2_D, the heterodimeric complex VDR/retinoid X receptor (RXR) is formed. This complex then engages co-repressors and co-activators to regulate gene transcription [25]. 1,25(OH)_2_D is known to play a crucial role in calcium homeostasis, as well as in controlling blood pressure and electrolytes. Through VDRs, 1,25(OH)_2_D controls genes associated with energy metabolism, detoxification, immune function, and the maintenance of calcium homeostasis. Therefore, those cells that express VDRs are direct targets for vitamin D [3].

As mentioned earlier, immune system cells such as dendritic cells, macrophages, B cells, and T cells are among those that express the VDR. While increasing the actions of regulatory T cells and interleukin-10 (IL-10), vitamin D, i.e., 1,25(OH)_2_D, inhibits the proliferation of T cells and the production of both interleukin-2 (IL-2) and interferon-gamma (IFN-γ). Vitamin D has immunoregulatory properties and therefore it could potentially repress some of the immune-mediated conditions, including inflammatory bowel disease, diabetes mellitus, and multiple sclerosis (MS) [26].

### 3.2. Vitamin D and Chronic Obstructive Pulmonary Disease

The underlying mechanisms of chronic obstructive pulmonary disease (COPD) are primarily characterized by oxidative stress, inflammation, and an imbalance between proteases and antiproteases [27]. Numerous studies have confirmed that COPD patients frequently have a vitamin D deficiency, which is a well-established risk factor for upper respiratory tract infections, and the latter is known as a significant COPD exacerbation factor [28]. Therefore, it is estimated that supplementation with vitamin D could potentially lower exacerbation rates in COPD or asthma patients who are suffering from a vitamin D deficiency, leading to the decreased incidence of acute respiratory tract infections. According to the study of Martineau et al. [28], vitamin D supplementation for a period of one year resulted in protective effects against moderate or severe exacerbation in participants with baseline serum 25(OH)D levels lower than 50 nmol/L, but without significant effects in those patients with COPD that had baseline serum 25(OH)D levels greater than 50 nmol/L. Based on the results obtained, it was concluded that treating vitamin D deficiencies in patients with COPD could lower the incidence of moderate or severe exacerbations [28].

Li et al. [27] performed a meta-analysis involving 2670 individuals diagnosed with COPD to better elucidate the impact of vitamin D supplementation on COPD. As per their findings, vitamin D supplementation in COPD patients resulted in the improvement of several studied parameters, including 6 min walk distance and pulmonary function [27]. Another study, conducted by Donastin et al. [29], also explored the effects of vitamin D supplementation on COPD. In this study, both studied groups (the control group and the treatment group) received vitamin D supplementation (1000 and 5000 IU/day, respectively) for three months and the applied supplementation resulted in the reduction of oxidative stress in both groups, together with significant enhancements in exercise tolerance and quality of life [29]. 

The beneficial impacts of vitamin D supplementation on COPD were confirmed by several other studies [30,31]; however, additional research is needed to better elucidate the effects of vitamin D in COPD patients and to define the supplement dosage that could lead to best results without the risk of potential toxic effects [32].

### 3.3. Vitamin D and the Coronavirus Disease 2019

The Coronavirus Disease 2019 (COVID-19) pandemic has increased the popularity, interest, and attention on vitamin D [33,34,35,36], although its benefits have been widely known for a long time. There are numerous identified risk factors that may increase the chances of COVID-19 development, such as male sex, obesity, diabetes, hypertension, microbiome status, and nutritional status [37]. It is also well-established that more serious COVID-19 infections may occur in those individuals that are immunocompromised due to their insufficient immune response. Contrarily, an immune response that is too strong, leading to “a cytokine storm”, could also result from severe COVID-19 infection in individuals that are not immunocompromised [37].

Several studies have indicated a potential link between insufficient serum vitamin D levels and an increased susceptibility to respiratory infections [36,38,39,40], including COVID-19 [36,41], while increased doses of this vitamin might help alleviate the severity of COVID-19 infections by decreasing inflammation and enhancing pulmonary function [41,42,43].

Some smaller studies have also indicated that taking vitamin D supplements could potentially provide a protective impact against COVID-19. These studies were, however, criticized for various limitations such as small sample size, lack of randomization, and potential confounding factors [35,44,45,46,47,48,49,50].

Vitamin D has the ability to stimulate the expression of angiotensin-converting enzyme 2 (ACE2) and influence the immune system through various mechanisms [36]. The ileum and kidneys have the highest levels of ACE2 expression, followed by the adipose tissue, heart, brainstem, lungs, circulatory system, stomach, liver, and nasal and oral mucosa [51]. The essential component for SARS-CoV-2 entrance into the host cell is ACE2. It is believed that clinical symptoms like respiratory distress, acute kidney injury, liver failure, and diarrhea are at least partially linked to the expression of ACE2 in these tissues because ACE2 is widely distributed in the lungs, heart, kidneys, inner lining of intestinal epithelial cells, and endothelium [51]. 

According to the study of Borsche et al. [52], the lowest threshold for vitamin D considered healthy should be roughly 125 nmol/L or 50 ng/mL. According to the authors, those serum levels of vitamin D could have numerous positive effects when it comes to COVID-19 infection and could potentially save human lives [52]. Grant et al. [53] suggest that people prone to the development of flu and/or COVID-19 infection should contemplate taking a daily dose of 10,000 IU of vitamin D_3_ for a few weeks initially, after which they should continue to take 5000 IU daily, to help lower their chances of infection. The authors further propose that even higher dosages could be useful in the treatment of those individuals who are already infected with COVID-19. However, they emphasize that large high-quality randomized controlled trials are required to properly estimate those recommendations [53]. 

When it comes to leading healthcare organizations, the World Health Organization (WHO) and the National Institutes of Health (NIH) currently recommend vitamin D supplementation for certain populations at risk of deficiency; however, neither organization currently recommends vitamin D supplementation as a specific measure for preventing or treating COVID-19. 

### 3.4. Vitamin D and Polycystic Ovary Syndrome 

More than 10% of couples experience infertility, and it is ranked among the most significant global public health challenges today. Polycystic ovary syndrome (PCOS) is responsible for anovulatory infertility in approximately 15% of women in their reproductive age. The etiopathogenesis and clinical symptoms of PCOS are significantly influenced by macronutrients and micronutrients, particularly vitamin D [54].

It is already well-established that PCOS patients have lower serum levels of vitamin D when compared to the general population [54]. According to Thompson et al. [55], as much as 67–85% of women diagnosed with PCOS have serum vitamin D levels lower than 20 ng/mL. These data suggest a possible association between serum vitamin D levels and PCOS. Furthermore, low serum vitamin D levels might increase the chances for the development of cardiovascular diseases (CVDs) in women with PCOS, and impact negatively on the symptoms of the disease, including insulin resistance, ovulatory dysfunction, infertility, irregular periods, hyperandrogenism, and obesity [54]. 

Vitamin D can positively impact the symptoms of PCOS through a number of different mechanisms of action. However, its impact on women with PCOS could vary due to genetic variability, different epigenetic mechanisms involved, and numerous disease phenotypes [56]. Therefore, we cannot expect vitamin D supplementation to exert the same effects in all women dealing with PCOS. For now, it is known that vitamin D could impact molecules participating in oxidative stress reactions at the follicular level, reduce hyperandrogenism, and have positive effects on serum levels of parathyroid and anti-Mullerian hormones [56]. Together with the aforementioned findings, it could also have positive effects on insulin resistance and obesity [56].

The randomized controlled trial performed by Fatemi et al. [57] aimed to examine the possible impacts of a combined supplementation with vitamins E and D on the number and quality of oocytes and embryos, as well as on the pregnancy rate in women diagnosed with PCOS. According to the results, a combined supplementation with vitamins E and D (400 mg/day and 50,000 IU/once in two weeks, respectively) for a period of 8 weeks led to significant improvements in clinical fertility in women with PCOS [57]. According to the authors themselves, further research is needed to better understand the impact of vitamin E + D supplementation on intra-cytoplasmic sperm injection outcomes and to evaluate the potentially different outcomes of the different dosages applied [57]. Another study conducted by Irani and Merhi [58] also confirmed beneficial outcomes of vitamin D supplementation in women with PCOS. To be more specific, in the mentioned study, vitamin D supplementation with 50,000 IU/week, for a total of 8 weeks, led to elevated levels of the soluble receptor for AGEs and decreased levels of serum anti-Mullerian hormone in vitamin D deficient women with PCOS, but no such effects were observed in vitamin D deficient women who did not have PCOS [58].

Despite the studies confirming the beneficial impacts of vitamin D supplementation on patients with PCOS, it is imperative to conduct further high-quality research with larger sample sizes and quality design to provide a clearer understanding of the influence of vitamin D supplementation within this specific population group and to define the adequate dosage able to exert therapeutic effects.

### 3.5. Vitamin D and Bone Health

Except for the potential connection to respiratory diseases and PCOS, there are other reasons why vitamin D has grown in popularity in recent years and why the awareness of its importance and the need to ensure adequate levels in the body has increased. Research has shown that many people, especially those inhabiting the northern parts of the world, are deficient in vitamin D [59]. This deficiency can result in detrimental effects on human bone health [60]. Inadequate levels of 25(OH)D in the bloodstream were linked to a higher likelihood of experiencing bone fractures and the decline in bone mineral density [61]. 

Vitamin D is widely recognized for its essential function in promoting calcium absorption and bone mineralization, with inadequate levels of vitamin D in the bloodstream potentially leading to inadequate calcium levels, consequently resulting in impaired bone health [62]. Because of the close connection and intertwined roles of calcium and vitamin D, the combined supplementation of these two essential nutrients (1200 mg of calcium and 800 IU of vitamin D per day) is usually recommended for preventing bone fractures in elderly individuals residing in institutions or those with established low serum levels of vitamin D [62].

However, the results of studies focusing on whether calcium and vitamin D supplements should be applied for the prevention and management of osteoporosis are still inconsistent. The findings from Weaver et al.’s meta-analysis [63] indicated that when adults received a combination of calcium and vitamin D supplements, it resulted in a 15% decrease in the risk of total fractures and a 30% decrease in the risk of hip fractures. Contrary to the mentioned study, Zhao et al.’s meta-analysis [64] did not observe any link between the use of calcium, vitamin D, or their combination and the risk of bone fractures in the population of older adults living in the community.

Some of the potential reasons why certain studies failed to find the connection between vitamin D supplementation and the risk of bone fractures could include the administration of low vitamin D doses (e.g., 10–20 µg/day), the very nature of the vitamin D supplement, the short duration of the studies, and the selection of the study group [65,66]. 

Further high-quality randomized controlled trials are essential to elucidate the impacts of vitamin D supplementation (with and without the administration of calcium) in preventing and treating bone fractures and osteoporosis in different population groups. For now, supplementation with vitamin D should not be prescribed routinely and should always be monitored by a healthcare professional to prevent potential toxic effects related to high doses (e.g., increased fractures and bone loss) [62,66].

### 3.6. Vitamin D and Cancer

Some research studies have suggested that vitamin D might have a potential role in preventing certain cancer types, including breast, colon, ovarian, bladder, lung, and prostate cancer [67,68], with several studies confirming that those with higher serum vitamin D levels have a reduced likelihood of developing certain malignancies [69,70,71,72]. 

It is believed that vitamin D exerts its anti-cancer properties through several different mechanisms, including promoting cell differentiation, reducing cell proliferation, and inhibiting the formation of new blood vessels that tumors need to grow [67]. However, the scientific data also imply that cancer cells could protect themselves from vitamin D’s anti-cancer properties by decreasing the cellular levels and function of calcitriol [20].

Keum et al. [73] found that taking 400–2000 IU/day of vitamin D reduced cancer-related deaths by roughly 13% among elderly individuals, although it did not significantly affect cancer incidence. On the other hand, when large-bolus dosages were administered infrequently, the same impact was not observed [73].

Some research studies have failed to find a significant association between vitamin D levels and cancer risk, and some randomized controlled trials (widely regarded as the highest standard in research) did not identify a substantial impact on both cancer incidence [67] and mortality rates [74].

The available evidence regarding the connection between vitamin D and cancer is inconclusive and further research is required to gain a comprehensive understanding of how vitamin D influences cancer prevention and treatment. While maintaining adequate vitamin D levels is important for overall health, it is noteworthy that taking vitamin D supplements is not recommended as a primary method for cancer prevention or treatment. It should certainly be emphasized that individuals who are concerned about their risk of cancer should consult their healthcare provider about screening and prevention strategies.

Additionally, it is of the utmost importance to emphasize that supplementing with vitamin D is not a cure that can eliminate cancer or take the place of leading a healthy lifestyle. It is crucial to maintain a positive environment and encourage a healthy way of living, which includes regular exercise and eating nutrient-dense foods. Both are the greatest available preventative methods and have been shown to affect overall health.

### 3.7. Vitamin D and Mental Health

Previously published studies have indicated that sufficient levels of vitamin D might have a positive impact on mental health [75,76], particularly in reducing symptoms of depression [76]. Our brain contains receptors for vitamin D, and it is believed that vitamin D could have a role in controlling mood and brain function [77,78,79,80,81]. 

Several observational studies have identified a correlation between lower serum vitamin D levels and an increased risk of depression [82,83]. Studies also found that individuals with depression had a higher probability of having insufficient vitamin D levels when compared to those without depression [84,85,86,87,88]. 

Other studies have found that taking vitamin D supplements may improve mood and reduce symptoms of depression in vitamin D deficient individuals [89]. On the contrary, certain research studies failed to find a significant correlation between serum vitamin D levels and depression, and the results of randomized controlled studies focused on investigating the effects of vitamin D supplementation on depression have been inconsistent [90,91].

Together with depression, a few other mental health conditions are believed to be linked to vitamin D deficiency, including schizophrenia [92], bipolar disorder [93], and attention-deficit/hyperactivity disorder (ADHD) [94]. Even though lower serum vitamin D levels have been found in people with serious mental disorders, when compared to healthy controls, it must be taken into account that people suffering from mental health conditions, such as schizophrenia, generally have poorer dietary habits, spend less time performing any type of physical activity, smoke more, and have an increased risk for the development of different types of diseases, all of which could be associated with reduced serum vitamin D levels [92]. Thus, the precise cause–effect connection between serum vitamin D levels and certain mental health conditions remains unclear, indicating the need for further research [92]. 

Furthermore, certain studies focused on investigating the effectiveness of taking vitamin D supplements as an add-on therapy for schizophrenia in adults. In an eight-week randomized, double-blind placebo-controlled study, vitamin D application at a dose of 14,000 IU per week resulted in significant improvements in cognition, but without significant impacts on mood, psychosis, and studied metabolic parameters, including body mass index, waist circumference, blood pressure, blood glucose, blood lipids, and HbA1c [95].

Contrary to the study of Krivoy et al. [95], Ghadery et al. [96] found that when compared to patients receiving a placebo, patients with schizophrenia who received a combined supplementation with vitamin D and probiotics for 12 weeks significantly improved general and total scores of the Positive and Negative Syndrome Scale (PANSS), together with ameliorations in serum levels of fasting glucose, total cholesterol, triglycerides, and several other studied parameters. 

Randomized controlled trials aimed at investigating the impact of supplementation with vitamin D on different mental health conditions are scarce, with most of the currently available studies having insufficient sample sizes and being performed in relatively brief timeframes, indicating a need for further high-quality research to fully comprehend and clarify the role of vitamin D in mental health.

Although potentially beneficial, vitamin D should not be used as a substitute for professional mental health treatment but as an add-on therapy based on the recommendation of a healthcare professional. Individuals who are experiencing symptoms of depression or other mental health conditions need to consult their healthcare provider regarding appropriate treatment options.

### 3.8. Vitamin D, Immune System, and Autoimmune Diseases

In addition to mental health, research has indicated that vitamin D has a significant role in immune system health in general [97]. It has been proposed that vitamin D deficiency could potentially be associated with a higher vulnerability to infections and autoimmune diseases [98]. Vitamin D helps to regulate the response of our immune system by promoting the generation and activity of immune cells such as T cells and B cells, which help to identify and destroy invading pathogens like bacteria and viruses [98].

Vitamin D also helps to reduce inflammation, which is a fundamental part of our immune system response. Chronic inflammation can lead to a range of health problems, including autoimmune disorders, so managing inflammation is important for maintaining good health. Nowadays, numerous research projects are aimed at determining how vitamin D affects the development of different autoimmune diseases, including MS and Hashimoto’s thyroiditis (HT). There are 2.8 million MS patients worldwide, and women are four times more likely than men to develop the illness. Young individuals are primarily affected by it, and it causes severe disability and a lower quality of life [99].

When speaking more specifically about the connection between vitamin D and MS, it is well-known that inadequate serum vitamin D levels are linked to the higher likelihood of developing MS [100,101], primarily because of its role in regulating the immune system [100]. Except for environmental risk factors, such as vitamin D status, numerous other factors were found to play a role in MS susceptibility, including lifestyle habits (e.g., diet and cigarette smoking) and genetics [102,103].

To date, the relationship between several vitamin D-related genes (specific genes with a role in encoding proteins engaged in the transport, metabolism, and function of vitamin D) and MS susceptibility has been widely studied; however, the results of the performed research are still inconclusive and do not allow for definite conclusions [101]. Despite the great effort of the research, the complex pathophysiology of MS makes it hard to develop successful strategies for treating symptoms and preventing disease progression [104].

In recent years, as low vitamin D levels were recognized as a risk factor for the development of MS, some clinical trials investigated the impact of taking vitamin D supplements for treatment and relapse prevention in people with MS [105], indicating how important it is to incorporate vitamin D supplementation in MS treatment protocols [106,107]. According to the research of Simpson-Yap et al. [108], MS patients receiving vitamin D at a daily amount of more than 5000 IU had a better quality of life. Similar findings were found in the research conducted by Beckmann et al. [109], where an oral supplementation of vitamin D for 12 months led to positive impacts on quality of life, particularly fatigue, in MS patients, the vast majority of whom were vitamin D deficient.

However, the existing data on the appropriate vitamin D dose and the treatment duration are limited and further high-quality randomized controlled trials are essential for us to be able to confirm the most successful strategy for vitamin D supplementation in MS individuals without leading to toxic side-effects [105,107,110].

Numerous studies conducted over the past few decades investigated the role of vitamin D in autoimmune thyroid diseases (AITD), including HT and Grave’s disease (GD). Like other human autoimmune diseases, the incidence of those related to the thyroid gland has also been associated with genetic predispositions and different environmental factors, including inadequate serum levels of certain nutrients, one of which is vitamin D [111]. The aforementioned finding was confirmed by studies conducted both on adults [112] and children [113] diagnosed with AITD.

Except for inadequate serum vitamin D levels in AITD patients, researchers managed to find an inverse relationship between serum vitamin D levels and the levels of anti-thyroid antibodies [114,115], indicating that deficiency in vitamin D could be involved in the pathogenesis of AITD. Although the correlation between inadequate serum vitamin D levels and the presence of AITD likely exists, it remains unclear whether inadequate vitamin D levels precede or result from AITD [111], leaving it to future research to clarify the exact cause-and-effect relationship.

When speaking about the immune system in general, a connection between a lack of vitamin D in the body and increased susceptibility to respiratory infections, including the flu and the common cold, has been established [116]. Some studies also indicate that vitamin D supplementation may reduce the risk of respiratory infections in certain populations, such as in people who are vitamin D deficient or who have compromised immune systems.

While vitamin D is a critical nutrient for immune health, it is just one piece of the puzzle, and maintaining a healthy immune system requires a balanced diet, regular exercise, adequate sleep, and other healthy lifestyle habits. There is a whole series of nutrients that are connected or have benefits for our immune system and that we all need to provide to our body through proper nutrition. Hence, it is very difficult to attribute benefits to just one nutrient and to exclude the influence or synergy of others.

For example, vitamin C is another nutrient that is important for immune system health [117]. Vitamin C and vitamin D are known to have different mechanisms of action in the body and they may complement each other in supporting a healthy immune system. Vitamin C is often recommended for preventing and treating colds and other respiratory infections. Vitamin C is also important for reducing oxidative stress and inflammation. Although important, the advantages of vitamin C are oftentimes (especially during COVID-19) slightly overstated and may not be fully supported by scientific evidence.

Scientific research and health information must always be approached with a critical eye and people must rely on evidence-based recommendations from reputable sources. While there may be trends or fads in the scientific literature, the consensus on the importance of adequate vitamin intake for supporting a healthy immune system is well-documented and supported by numerous research projects. In conclusion, ensuring adequate vitamin levels is of critical importance for supporting immune function, but there is no one nutrient that can prevent or cure infections.

### 3.9. Vitamin D and Diabetes

In recent years, the relationship between vitamin D and diabetes has been gaining more and more attention. Although the two types of diabetes, type 1 diabetes (T1D) (autoimmune disease preventing the pancreas from making insulin) and type 2 diabetes (T2D) (a chronic condition characterized by the body’s inability to use insulin properly) have completely different etiologies, vitamin D is believed to have a crucial role in the pathophysiology of both the aforementioned types of the disease [118,119].

There is an increasing body of scientific evidence supporting the role of vitamin D in pancreatic beta cell function, systematic inflammation related to T2D, and insulin sensitivity [120], and numerous studies confirmed that elevated levels of vitamin D in the bloodstream are linked to a reduced risk of developing T2D [121,122,123], indicating that it may also have a role in the occurrence of T2D complications [43,124].

Similar to the connection between serum vitamin D status and other health conditions mentioned in previous subsections, the results confirming the correlation between inadequate serum vitamin D levels and T2D were mainly obtained from observational studies; thus, we cannot draw cause–effect conclusions. Several systematic reviews and meta-analyses aimed to elucidate whether taking vitamin D supplements could have positive effects in preventing and treating T2D. According to the results of a systematic review performed by Li et al. [125], vitamin D supplementation positively influenced serum levels of 25(OH)D and reduced insulin resistance; however, without an impact on HbA1c, fasting glucose level, and fasting insulin level. Another meta-analysis performed by Lee et al. [126] found no difference in fasting blood glucose levels and modest improvements in HbA1c after vitamin D supplementation in adults with T2D. Some other studies also showed promising results in people with prediabetes [127,128]; however, further research studies need to be performed to confirm the appropriate dose of vitamin D supplements and the impact on isolated population groups (e.g., people with prediabetes with and without obesity).

The findings of research studies investigating the link between vitamin D deficiency and the increased risk for the development of T1D are also inconclusive. In the Mendelian randomization study conducted by Manousaki et al. [129], inadequate serum levels of vitamin D were not linked to a higher risk for T1D. The results from the meta-analysis performed by Najjar et al. [130] have not shown an association between T1D risk and selected gene variants related to serum 25(OH)D concentrations. Contrarily, the meta-analysis conducted by Hou et al. [131] led to a conclusion that serum levels of vitamin D are significantly inversely related to the susceptibility for T1D development.

There are a limited number of research projects dealing with the impact of vitamin D supplementation on T1D treatment. Based on the results of their systematic review, Nascimento et al. [132] suggest that vitamin D supplementation could potentially positively influence glycemic control in children and adolescents with T1D; however, it is important to highlight that the studies included had a questionable methodological quality and that further research is of importance for the determination of the exact effects of vitamin D supplementation as an adjunctive therapy in treating this specific population group. The study of Bogdanou et al. [133] also showed promising results. In the aforementioned 3-month randomized, double-blind, placebo-controlled study, patients with T1D received 4000 IU/day of vitamin D_3_ or a placebo and according to the obtained results, HbA1c was significantly ameliorated in the intervention group when compared to the controls.

Further high-quality randomized controlled trials and meta-analyses are essential to confirm the impact of vitamin D supplementation in T1D treatment and to define adequate doses for different population subgroups. 

### 3.10. Vitamin D and Cardiovascular Disease

CVDs are primary causes of morbidity and mortality worldwide. The deficiency of vitamin D has been identified as one of the risk factors for CVD development and its potential protective role in CVD prevention has been widely studied. The discovery of a nuclear vitamin D receptor on cardiac muscle cells and vascular endothelial cells led to the assumption that the deficiency of vitamin D could influence the development of CVDs [134].

Until today, low serum vitamin D levels have been linked to an increased susceptibility for the development of coronary artery disease, heart failure, atrial fibrillation, hypertension, chronic kidney diseases, and several other conditions related to cardiovascular health [134,135]. Similar to the association with all the other health conditions discussed in previous subsections, the results examining the link between vitamin D and CVDs are mixed and further research is needed for us to be able to draw cause–effect conclusions.

The number of randomized controlled trials aimed at investigating the influence of vitamin D supplementation on CVD events and treatment is rising; however, for now, the results of published and available studies are inconsistent. In the meta-analysis performed by Ford et al. [136], the findings of a sensitivity analysis indicated that taking vitamin D supplements, in comparison to controls not taking vitamin D, significantly reduced cardiac failure events, but did not influence stroke and myocardial infarction. Another meta-analysis failed to find the connection between vitamin D supplementation and both systolic and diastolic blood pressure in children and adolescents [137]. Similarly, Barbarawi et al.’s [138] meta-analysis, including 83,291 patients, concluded that supplementation with vitamin D was not linked to a decline in cardiovascular events, CVD mortality, stroke, myocardial infarction, or both CVD and all-cause mortality. On the other hand, supplementation with vitamin D was found to be beneficial in reducing bloodstream levels of total cholesterol, LDL cholesterol, and triglycerides, with a more noticeable effect observed in patients who were vitamin D deficient [139].

Further randomized controlled trials with adequate power, sample size, and methodology are needed for us to make conclusions about the exact impact vitamin D supplementation has on CVDs when it comes to prevention and treatment.

## 4. Shedding Light on the Global Pandemic of Vitamin D Deficiency—Do We Need Urgent Action?

Today, vitamin D deficiency is a serious public health problem present in approximately 50% of the global population, independent of age and ethnicity [36,140]. The results of a meta-analysis conducted by Garland et al. [141] found that inadequate serum vitamin D levels are associated with a higher susceptibility for all-cause mortality. Because of those findings, other researchers were prompted to suggest measures for preventing vitamin D deficiency in the global population. These measures were mainly focused on raising the currently recommended vitamin D intake levels [142], together with promoting fortified products and supplementation [143]. 

One reason for the widespread deficiency is lifestyle changes, with people now spending more time indoors and consequently not receiving adequate sunlight exposure, which is a crucial source of vitamin D [144]. Additionally, certain factors such as skin color, age, and geographic location can influence an individual’s capacity to generate vitamin D through sun exposure [145].

Today, the global population is experiencing an unprecedented increase in the proportion of elderly individuals. As individuals grow older, their skin becomes less effective at synthesizing vitamin D through sun exposure, which can increase the risk of deficiency. 

Along with the aforementioned findings, dietary factors such as the limited intake of vitamin D-rich foods, like fatty fish and fortified dairy products, can also contribute to vitamin D deficiency [3,146]. Therefore, following rigorous vegetarian or vegan diets needs to be supervised by a nutritionist to prevent vitamin D and other deficiencies.

The consequences of vitamin D deficiency could be serious, including an elevated risk of bone fractures, muscle weakness, weakened immune system [147], diabetes, heart disease, and certain types of cancer, as discussed in Section 3. In the opinion of the authors, potentially because of all the reasons mentioned, physicians are now more aware of the significance of testing for vitamin D deficiency and vitamin D supplementation. In addition to awareness, the progress of technology certainly plays an important role as well. Nowadays, testing for vitamin D is performed routinely in most laboratories.

## 5. Is It Possible to Obtain Sufficient Amounts of Vitamin D from Your Diet?

There are several natural food sources of vitamin D, and some with the highest content include the following: fatty fish (salmon, mackerel, tuna), egg yolk, beef liver (it is important to consume it in moderation due to its high vitamin A content), and mushrooms (especially shiitake mushrooms). When sterols such as ergosterol (a precursor for vitamin D_2_) in fungi or the direct precursor for cholesterol, 7-dehydrocholesterol (a precursor for vitamin D_3_), found in animals and phytoplankton are subjected to UVB radiation, they undergo a non-enzymatic conversion into vitamin D_2_ and vitamin D_3_, respectively [3].

Other important food sources of vitamin D are fortified foods such as milk and dairy products (yogurt, cheese), breakfast cereals, orange juice, plant-based milks (soy milk, almond milk), margarine, and tofu (Table 1). The labels of packaged foods need to be checked to verify vitamin D fortification [146,148]. 

Even though some products commonly consumed as a part of the human’s diet are fortified, there is still a limited number of food sources of vitamin D. Therefore, even when the richest sources are being consumed in manageable amounts, it makes it hard to reach recommended daily intake levels of vitamin D [12].

One randomized controlled trial conducted in Dutch older adults estimated that the mean daily vitamin D intake was 4.3 µg, as assessed through a food frequency questionnaire. In the mentioned study, margarine and butter were the primary sources contributing to the overall vitamin D intake [150]. An average vitamin D intake lower than the recommended dietary allowance was confirmed in numerous other studies conducted in different population groups [151,152,153,154,155]. 

What is more, the amount of vitamin D in natural (animal) food sources can vary depending on numerous factors, e.g., the vitamin D content in the food the animal in question has been fed with [156].

Fortified foods should not be relied upon as the sole source of vitamin D for people. Vitamin D should also be received from sunlight exposure, natural food sources, or dietary supplements. The necessary amount of vitamin D may vary based on several factors, including sex, age, health status (presence of different diseases, including kidney disease), and sun exposure [15].

The WHO currently recommends the following daily doses of vitamin D for different population groups: infants and children under 5 years old, 400–1000 IU/day; pregnant and breastfeeding women, 600–800 IU/day; adults aged 18–50 years old, 600–800 IU/day; and adults over 50 years old, 800–1000 IU/day [145]. IU represents a measure of the biological activity of a substance, such as a vitamin or hormone. For vitamin D, IU is a measure of its potency or strength. When a recommended daily intake of vitamin D is expressed in IU/day, it represents the amount of vitamin D recommended for individuals to consume each day to meet their needs. For example, if the recommended dose of vitamin D for a specific population group is 800 IU/day, it means that they should aim to consume 800 IU of vitamin D from various food sources, including supplements if needed.

The conversion of IU to grams or milligrams can vary based on the particular nutrient and the form of the supplement. It is always best to follow the dosage instructions on the supplement label or to consult a healthcare provider, especially knowing that the absorption of vitamin D can be influenced by several factors that should be considered when defining the necessary doses.

Singh and Bonham [157] provided a formula for the calculation of the dose in IU that takes into account the individual’s body mass index. It is expressed as follows [157]:‘Dose in IU = [(8.52—desired change in 25(OH)D) + (0.07 × age) − (0.20 × body mass index)
+ (1.74 × serum albumin) − (0.62 × starting 25[OH]D concentration)]/(−0.002)’

They theorized that the increased requirements for higher vitamin D doses, such as in nursing home residents, might be attributed to insufficient sun exposure. Nonetheless, their assessments determined that age alone did not significantly impact the response to vitamin D treatment. They acknowledged that numerous patients necessitate ongoing therapy over an extended period of time. Additionally, they noted that a dose of 5000 IU/day is typically required to correct a deficiency, while a standard maintenance dosage should be no less than 2000 IU/day [157].

The above-mentioned recommendations could potentially be applied to older people, because their skin is less efficient at producing vitamin D in response to sunlight. Also, older adults often spend more time indoors or cover their skin more, which can further reduce their vitamin D production. Hence, older adults are at a higher risk for vitamin D deficiency and might find vitamin D supplementation beneficial. The National Institute on Aging (NIA) and WHO also recommend that adults aged 70 or older should consider taking a vitamin D supplement in the dose of 800–1000 IU/day [146].

People of all generations with dark skin can also have the discussed deficiency. Melanin, the pigment that gives skin its color, can diminish the skin’s capacity to generate vitamin D in response to sunlight [158]. As a result, individuals with darker skin may need increased sunlight exposure or higher doses of vitamin D supplements to sustain adequate vitamin D levels [15,159]. 

Similar to older people, infants could also be vitamin D deficient, especially if breastfed (both exclusively or partially). A mother’s milk is the best source of nutrition for babies, but it typically does not contain enough vitamin D to meet the child’s needs [160]. The American Academy of Pediatrics advises that every infant, including those who are exclusively or partially breastfed, receive a vitamin D supplement in the dose of 400 IU/day from the first days of life until they transition to consuming at least one liter or quart of vitamin D-fortified formula or whole milk per day. This is recommended because infants who are breastfed might not get enough vitamin D from breast milk and sun exposure combined. Infants fed with formula typically obtain an adequate amount of vitamin D from the formula itself (when their intake reaches at least 1 liter or quart of vitamin D-fortified formula or whole milk per day), but it is important to check the food label to make sure that the formula is fortified with vitamin D. The supplementation should always be prescribed by a pediatrician and the prescribed dosage should not be exceeded [161].

Finally, people with certain medical conditions (e.g., Crohn’s disease or celiac disease) might experience challenges in absorbing vitamin D from their food [162]. It is also known that some other medical conditions (liver or kidney disease) can disrupt the absorption or processing of vitamin D. As a result of all the aforementioned findings, elderly individuals, infants, individuals with dark skin, or those with the above-mentioned medical conditions may need higher doses of vitamin D from food sources or supplements to maintain optimal levels.

It is important to highlight that vitamin D is a fat-soluble vitamin, meaning that it is stored in the body’s fat cells. Therefore, people with higher body fat and obesity might also need more vitamin D to maintain adequate serum levels in the body [163].

## 6. Can You Take Too Much of Vitamin D?

Taking vitamin D is generally considered safe if it is taken in recommended doses. However, too much of a good thing can sometimes do more harm than good. The uncontrolled intake of excessively high doses of vitamin D can have unwanted and very serious consequences. Intoxication with vitamin D is mainly caused by the excess intake of vitamin D through supplements [164]. Vitamin D toxicity is unlikely to be caused by excessive exposure to sunlight because our bodies can control the production of vitamin D produced by sunlight [164].

When it comes to dietary intake, as mentioned in Section 5, there are only a few natural food sources of vitamin D, and usually the estimated dietary intake of vitamin D is far below the recommended dietary allowance [165], which makes it almost impossible to ingest doses that could lead to toxic effects.

What is more, some people may have underlying health conditions that could be worsened by taking excessive doses of vitamin D. One of those conditions is hypercalcemia, characterized by high levels of calcium in the blood that can cause kidney stones, heart problems, and other serious health issues [166,167,168]. Also, people suffering from kidney disease have a reduced ability to process vitamin D, which can result in elevated levels of calcium in the bloodstream [169]. Similarly, people suffering from liver-related diseases might have difficulty processing excessive doses of vitamin D because vitamin D is processed by the liver [170].

Certain medications, such as steroids, can interfere with vitamin D metabolism and increase the likelihood of experiencing toxicity [171]. Some medical conditions, such as sarcoidosis or tuberculosis, can cause the body to produce too much vitamin D, again leading to toxicity [172]. Therefore, healthcare providers can help determine the appropriate dose of vitamin D based on individual needs and the health status of their patients.

Vitamin D supplements are readily accessible, and in 2020, the industry’s estimated market worth exceeded $1.1 billion, with a projected increase to nearly $1.6 billion by 2025. This reflects an annual growth rate of over 7%, driven by an increased focus on nutrition and overall health [173].

The widespread use of vitamin D supplements has, at least partially, been driven by public awareness campaigns highlighting the dangers of excessive sun exposure leading to skin cancer, the link between vitamin D deficiency and various chronic illnesses, and the connection between vitamin D levels and optimal immune system performance [173].

When deciding on vitamin D supplementation, there are several factors to consider. The first factor concerns the type of vitamin D. Vitamin D_3_ (cholecalciferol) is the recommended type of vitamin D because it is more efficient for increasing and sustaining blood levels of vitamin D when compared to vitamin D_2_ (ergocalciferol) [174]. The second factor concerns the dosage. The recommended daily vitamin D intake varies based on numerous factors, including age and sex [146]. The third factor concerns quality. The best supplements are supplements from reputable manufacturers that undergo third-party testing to ensure purity and potency [14]. The fourth factor concerns the form. There are various forms of vitamin D supplements, including capsules, tablets, gummies, and liquid drops. A form that is easy to take and that fits the lifestyle of the individual should be chosen [175,176]. The final factor concerns the absorption. Vitamin D is a fat-soluble vitamin, meaning it is more effectively absorbed when consumed alongside a meal containing a certain amount of dietary fat [177].

Several other nutrients are necessary for vitamin D to function properly in our body. These include calcium (vitamin D helps the absorption and use of calcium in the body, a crucial nutrient for building and maintaining healthy bones and teeth) [178], magnesium (required for vitamin D activation, and it helps to regulate calcium levels in the body) [179], phosphorus (another mineral critical for bone health, and it works together with calcium and vitamin D to keep bones strong) [180], vitamin K_2_ (helps to regulate calcium metabolism in the body, and it works together with vitamin D to support bone health) [181], zinc (mineral that participates in the functioning of the immune system, and it is needed for the proper metabolism of vitamin D) [182], and vitamin A (participates in controlling gene expression and cell differentiation, and it works together with vitamin D to support bone health) [183]. Therefore, a balanced diet that includes a variety of nutrient-rich foods must be upheld to maintain optimal health. 

## 7. Future Developments in Vitamin D Therapy and the Potential Role of Personalized Supplementation Plans

The existing epidemiological evidence suggests that vitamin D deficiency is prevalent on a global scale [184,185,186]. Most of the previously published studies that discussed the health risks and morbidity associated with vitamin D deficiency provided strong and well-rounded evidence [187,188,189]. It is important to recognize that there have been a few studies in the past decade that did not find positive effects of vitamin D [190,191,192]. In addition, some recent data from randomized controlled trials with vitamin D have presented conflicting outcomes. Nevertheless, the overall consensus from most of these studies, especially those with extended observation periods and specific outcome measures, indicate that vitamin D has positive effects in terms of cancer prevention and reducing overall mortality rates [187,189]. Therefore, in the future, greater consideration should be given to future developments in vitamin D therapy, i.e., potentially consider the possibilities of precision supplementation through personalized supplementation plans, as our understanding of individual variations in vitamin D metabolism and requirements improves. Precision supplementation and its practical implementation may vary based on available resources, technology, and scientific advancements. Regular blood tests can measure an individual’s vitamin D levels and provide an objective assessment of whether present vitamin D status is within a healthy range. Genetic factors (person’s genetic predisposition to vitamin D-related health issues), lifestyle choices (questions about sun exposure habits, dietary preferences, physical activity levels, and other relevant habits), and environmental influences (geographic location and seasonal changes in sunlight availability) should also be considered. Understanding why serum vitamin D concentrations fluctuate with seasons and how to maintain summertime levels during winter months is crucial, especially since there is an inverse correlation between these concentrations and the risk of CVDs and infections [193,194]. During the winter months, when solar UVB radiation is scarce, serum vitamin D levels tend to be around 50–70% lower than their peak values in the summer [195,196,197]. This drop is primarily due to vitamin D being stored in muscles, which is connected to the vitamin D levels in the bloodstream, and its movement to the bloodstream is controlled by the parathyroid hormone as necessary, especially during the winter when serum vitamin D levels decline [198,199,200]. 

In the general population with documented vitamin D deficiencies, cholecalciferol should be considered as the first choice for both prophylactic and treatment options. Calcifediol is the second option and should be considered when the use of cholecalciferol does not result in the improvement in serum vitamin D levels or when a fast increase in serum levels is necessary [185]. The dosage of both cholecalciferol and calcifediol should be determined based on serum vitamin D levels, chronological age, and, in the case of cholecalciferol, body weight as well [185]. Based on all the findings mentioned, healthcare providers can develop a personalized supplementation plan that might involve recommending specific vitamin D dosages, frequencies, and the duration of supplementation tailored to the individual’s unique needs [201]. As part of precision supplementation, individuals should also receive education about the importance of vitamin D, the way it affects their health, and the role of lifestyle factors in maintaining optimal levels. Increased knowledge regarding the significance of vitamin D and its sources might encourage more people to prioritize their vitamin D intake via a combination of safe sunlight exposure, diet, and supplementation, if necessary.

With the rise of digital health tools and wearable devices, individuals could monitor their sunlight exposure and other relevant metrics as has already been done for example for glucose, sport activities, etc. [202,203]. These data can be integrated into a precision supplementation plan for more accurate recommendations. 

Also, vitamin D therapy could be integrated with other treatments or therapies to enhance their effectiveness. For instance, combining vitamin D supplementation with certain cancer treatments, autoimmune disorder management, or metabolic syndrome interventions could lead to better outcomes [204]. In the future, researchers might develop much more effective and bioavailable forms of vitamin D supplements, potentially improving their absorption and utilization by the body. This could include new delivery methods, such as nanoparticles or liposomal formulations [205]. 

As our knowledge of vitamin D’s role in various health conditions expands from year to year, as this article shows, we might see its application in new areas. Research might lead to more refined guidelines for vitamin D intake and supplementation. Regulatory agencies and health organizations will be able to update their recommendations based on the latest scientific evidence. With the rise of digital health technologies, including wearable devices and remote monitoring tools, individuals could also receive more accurate insights into their vitamin D levels and overall health status. All of these could lead to better-informed decisions about vitamin D supplementation. 

## 8. Conclusions

Vitamin D is an essential nutrient playing an important role in promoting overall health and well-being. The association between vitamin D status and health is already well-established; however, the exact beneficial effects of vitamin D supplementation on the prevention and treatment of different health conditions and the determination of an appropriate dosage to exert those beneficial effects in various population groups are still inconclusive. Therefore, further studies are needed to provide more comprehensive insight into the topic and help us make more concrete conclusions. 

Nowadays, we are becoming more and more aware of the individual variations in vitamin D metabolism and requirements. Due to individual’s unique needs, the future might lie in personalized supplementation plans with the aim of maintaining good health and preventing and treating individual deficiency-related conditions. 

## Figures and Tables

**Figure 1 nutrients-16-01176-f001:**
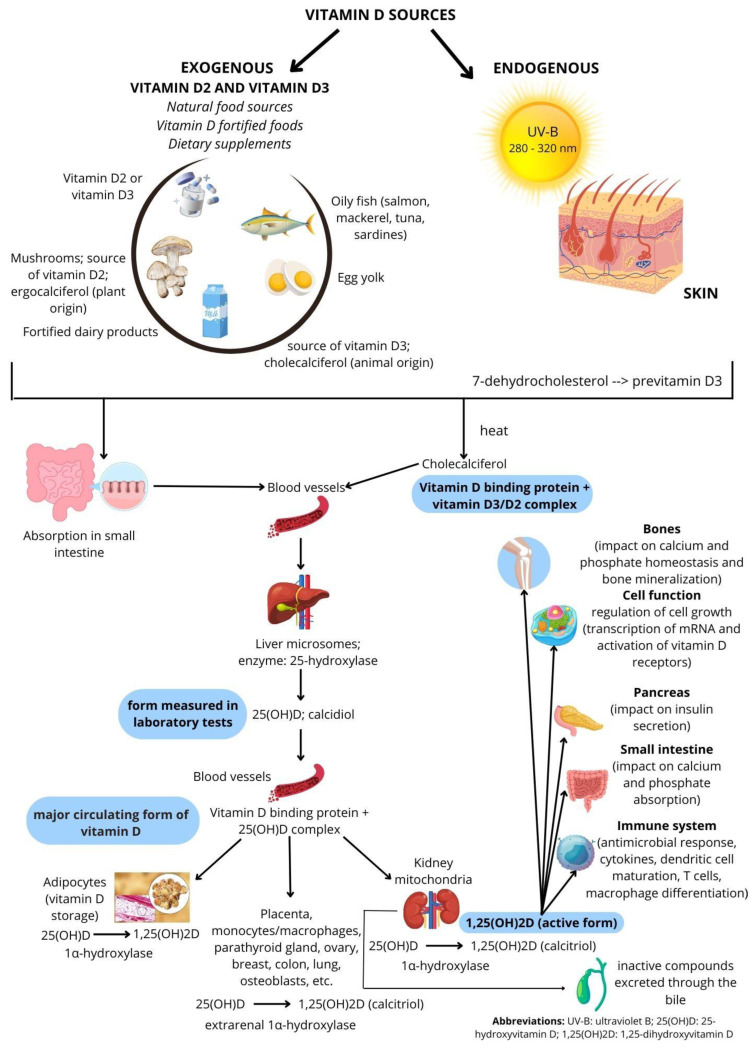
Vitamin D metabolism and pathway.

**Table 1 nutrients-16-01176-t001:** The amount of vitamin D in selected food sources [149].

Food Source	The Amount of Vitamin D (μg/100 g)
Fish oil, cod liver Fish, mackerel, Atlantic, raw Fish, swordfish, raw Fish, salmon. Pink, raw Fish oil, sardine Fish, sardines, canned Fish, trout, mixed species, raw Fish, tuna, raw Fish, shark, mixed species, raw Liver, beef Mushrooms, shiitake, raw Egg, yolk, raw, fresh Milk, low-fat (1% milk fat), with added vitamin D Yogurt, nonfat milk, plain Soy milk Orange juice, chilled, includes from concentrate, with added calcium and vitamin D Margarine, NFS Butter, NFS Cheese, cheddar Cheese, feta	250.0 16.1 13.9 10.9 8.3 4.8 3.9 1.7 0.6 1.2 0.5 5.5 1.0 1.2 0.7 1.0 3.7 0.4 0.6 0.4

## Data Availability

No new data were created or analyzed in this study. Data sharing is not applicable to this article.

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
