# Peer review of "The Power of Vitamin D: Is the Future in Precision Nutrition through Personalized Supplementation Plans?"

_nutrients, 2024, doi:10.3390/nu16081176_

Round 1

Reviewer 1 Report

Comments and Suggestions for Authors

Thank you for the opportunity to review this manuscript entitled “The Power of Vitamin D: Is the Future in Precision Nutrition Through Personalized Supplementation Plans?

Overall, this is an extremely well done review of vitamin D. The major two comments I have is: 

1) figure one vitamin D metabolism, and pathway needs to include extra renal conversion of 25 to 125D. This should be added in the discussion as well.

2) The second main point is the title needs to be more clearly addressed in the review. Section 7 addresses it but the title of does not capture it.

Section 7 “Future Developments in Vitamin D Therapy” 

might be retitled as “Future Developments in Vitamin D therapy and the Potential Role of Personalized Supplementation Plans” or something like that. Something that makes it much more clear that the question in the title is being addressed in a clear manner.

Minor comments, 

Page 4 of 27, section 3 

vitamin D, and its positive health effects

lines 122-124

….by preventing the invasion of harmful bacteria, inhibiting, inflammation, and preserving diet function, VDR is crucial in controlling intestinal homeostasis. Maybe “activation of VDR”

Page 4 of 27, Section 3.2 

Lines 151-153

But with no effect on upper respiratory infections, and patients with COPD, that had baseline serum, 25 hydroxy D levels “lower” than 50 nmol/L.

This should be no effect in patients with baseline levels “greater” than 50 nmol/L

Page 6 of 27 line 224

Should be levels lower “than” 20 ng/ml rather than “that”

Page 8 of 27, line 363

Spell out PANSS - Positive and negative syndrome scale (PANSS)

Page 9 of 27 line 388

Spell out MS, multiple sclerosis (MS)

Page 12, of 27, line 568

Instead of nowadays this analysis is performed routinely make it clear and say nowadays “testing for vitamin D” is performed routinely in most laboratories.

Page 13 of 27 line 615 

starts with WHO

Should be “The world health organization (WHO) currently recommends….

Page 15, line 675

…need more vitamin D to maintain adequate levels in the body.

I think this should be …..need more vitamin D to maintain adequate “serum levels” in the body.

Page 17 of 27 line 783

I think sport activates should be sport activities

Reviewer 2 Report

Comments and Suggestions for Authors

The peer-reviewed manuscript titled 'The Role of Vitamin D: Precision Nutrition Through Personalized Supplementation Plans' is a comprehensive study on the beneficial effects of vitamin D on various cells, organs, and systems of the body. Additionally, the authors present the findings of studies on the potential role of vitamin D and its supplementation in preventing and/or treating several diseases prevalent in a significant proportion of the global population. They acknowledge the discrepancies and ambiguities in the results obtained and suggest an individualised approach to vitamin D supplementation.  The manuscript's content justifies further research on the effectiveness of individualised vitamin D supplementation, while also highlighting the risks of overdosing.

The authors emphasise that vitamin D supplementation should only be considered after ensuring exposure to other sources of this vitamin, such as sunlight and food products. They also suggest the use of modern technology to tailor supply and supplementation to individual needs and abilities.

The manuscript has been prepared soundly and presents balanced and thoughtful statements.

However, it includes a significant proportion of older literature, i.e. from more than 10 years ago, despite being mostly based on current literature. Given the number of items in the References section and the abundance of vitamin D literature published in the last decade, it is recommended to review the References section and remove any redundant 'old' items that overlap in content with 'newer' items.
